# Echolocating toothed whales use ultra-fast echo-kinetic responses to track evasive prey

Heather Vance[1]*, Peter T Madsen[2], Natacha Aguilar de Soto[3], Danuta Maria Wisniewska[4], Michael Ladegaard[2], Sascha Hooker[1], Mark Johnson[5]*

[1]Sea Mammal Research Unit, University of St Andrews, St Andrews, United Kingdom; [2]Zoophysiology, Department of Biology, Aarhus University, Aarhus, Denmark; [3]BIOECOMAC, Department of Animal Biology, Edaphology and Geology, University of La Laguna, La Laguna, Spain; [4]Centre d'Etudes Biologiques de Chizé, CNRS, Villiers en Bois, France; [5]Aarhus Institute of Advanced Studies, Aarhus University, Aarhus, Denmark

**Abstract** Visual predators rely on fast-acting optokinetic responses to track and capture agile prey. Most toothed whales, however, rely on echolocation for hunting and have converged on biosonar clicking rates reaching 500 /s during prey pursuits. If echoes are processed on a click-by-click basis, as assumed, neural responses 100× faster than those in vision are required to keep pace with this information flow. Using high-resolution biologging of wild predator-prey interactions, we show that toothed whales adjust clicking rates to track prey movement within 50–200 ms of prey escape responses. Hypothesising that these stereotyped biosonar adjustments are elicited by sudden prey accelerations, we measured echo-kinetic responses from trained harbour porpoises to a moving target and found similar latencies. High biosonar sampling rates are, therefore, not supported by extreme speeds of neural processing and muscular responses. Instead, the neurokinetic response times in echolocation are similar to those of tracking responses in vision, suggesting a common neural underpinning.

*For correspondence:
hmv@st-andrews.ac.uk (HV);
markjohnson@bios.au.dk (MJ)

Competing interest: The authors declare that no competing interests exist.

## Introduction

Response speed critically determines the outcome of interactions between mobile prey and pursuit predators. Prey must react rapidly to survive while predators must counter evasive prey movements quickly to gain sustenance. The fitness implications of these interactions have led to an evolutionary escalation of response times with the fastest-responding individuals being the most likely to survive and reproduce (*Dawkins and Krebs, 1979*). However, sensory and motor requirements are asymmetric for predators and prey. Responding to close-approaching predators, prey may trade accuracy for speed, relying on imprecise ballistic motor actions triggered by strong sensory cues that require little neural processing (*Domenici and Batty, 1997*; *Turesson et al., 2009*). This has led to extremely fast responses employing short efferent pathways linking sensors directly to muscles, such as the Mauthner-cell-mediated C-start response of teleost fish to fluid motion from oncoming predators (*Eaton and Hackett, 1984*). C-start responses are characterised by sudden accelerations (*Domenici and Blake, 1997*) and unpredictable trajectories (*Moore and Biewener, 2015*), with response latencies as low as 5–10 ms (*Eaton et al., 1977*). In contrast, predators must sacrifice speed for accuracy, typically requiring greater sensory resolution and motor-planning capabilities to track and successfully pursue evasive prey. The increased processing needed to locate prey in complex natural scenes, along with the typically larger body size of predators, inevitably results in slower movement responses

**eLife digest** In the animal world, split-second decisions determine whether a predator eats, or its prey survives. There is a strong evolutionary advantage to fast reacting brains and bodies. For example, the eye muscles of hunting cheetahs must lock on to a gazelle and keep track of it, no matter how quickly or unpredictably it moves. In fact, in monkeys and primates, these muscles can react to sudden movements in as little as 50 milliseconds – faster than the blink of an eye. But what about animals that do not rely on vision to hunt?

To find food at night or in the deep ocean, whales and porpoises make short ultrasonic sounds, or 'clicks', and then listen for returning echoes. As they close in on a prey, they need to click faster to get quicker updates on its location. What is unclear is how fast they react to the echoes. Just before a kill, a harbour porpoise can click over 500 times a second: if they wait for the echo from one click before making the next one, they would need responses 100 times faster than human eyes.

Exploring this topic is difficult, as it requires tracking predator and prey at the same time. Vance et al. took up the challenge by building sound and movement recorders that attach to whales with suction cups. These were used on two different hunters: deep-diving beaked whales and shallow-hunting harbour porpoises. Both species adapted their click rate depending on how far they were from their prey, but their response times were similar to visual responses in monkeys and humans. This means that whales and porpoises do not act on each echo before clicking again: instead, they respond to groups of tens of clicks at a time. This suggests that their brains may be wired in much the same way as the ones of visual animals.

In the ocean, increased human activity creates a dangerous noise pollution that disrupts the delicate hunting mechanism of whales and porpoises. Better understanding how these animals find their food may therefore help conservation efforts.

---

compared to prey, and these are often partly offset by ingenious capture tactics (*Catania, 2009*; *Wiley et al., 2011*), cooperative hunting (*Domenici et al., 2000*; *Benoit-Bird and Au, 2009*), or sensory/cognitive superiority (*Bailey et al., 2012*).

For most macro-predators, binocular vision is the predominant sensory modality for hunting, and prey movements are tracked within complex visual scenes by a combination of smooth and stepped (saccadic) movements of the eye muscles that manipulate gaze direction and depth of view (*Land, 1999*). This dynamic tracking is achieved by a set of optokinetic responses in humans and other primates with latency of 50–250 ms (*Miles et al., 1986*; *Erkelens, 2006*; *Kirchner and Thorpe, 2006*) that have been described as ultra-fast (*Girard et al., 2008*).

The evolution of echolocation in bats and toothed whales has allowed erstwhile visual predators to occupy foraging niches with low light levels such as deep or murky waters or at night. In contrast with vision, which relies on exogenous continuous light energy, echolocation is a discrete-time active sense in which ultrasonic pulses are used to sample the environment. These intense sounds potentially offer an auditory cue to prey, and many insects targeted by bats have developed ultrasonic hearing provoking an acoustic arms race (*Goerlitz et al., 2010*). In contrast, very few marine organisms have ultrasonic hearing (*Wilson et al., 2007*). This allows echolocating toothed whales in dark waters to approach most prey without being detected until close enough that prey can sense them via hydro-mechanical cues (*Wilson et al., 2018*). Such short-range detection necessitates rapid responses from prey which must be countered by fast biosonar-informed locomotory adjustments if toothed whales are to capture agile prey.

Echolocators control information flow, in the form of returning echoes, by the rate at which sound transients are produced to probe the environment. Both bats and toothed whales appear to avoid ambiguous echo information by adjusting their sampling interval, effectively expanding and contracting the acoustic depth of field, to accommodate changes in the two-way acoustic travel time as they approach targets (*Madsen and Surlykke, 2013*; *Stidsholt et al., 2021*). Clicks are accordingly produced slowly for long-range echolocation (long sensing range), but more rapidly and with lower intensity (short sensing range) for tracking nearby targets (*Griffin, 1958*; *Wisniewska et al., 2014*; *Salles et al., 2020*). In a remarkable convergence, toothed whales and bats both use rapidly accelerating series of clicks, known as buzzes, when approaching prey (*Madsen and Surlykke, 2013*), thereby

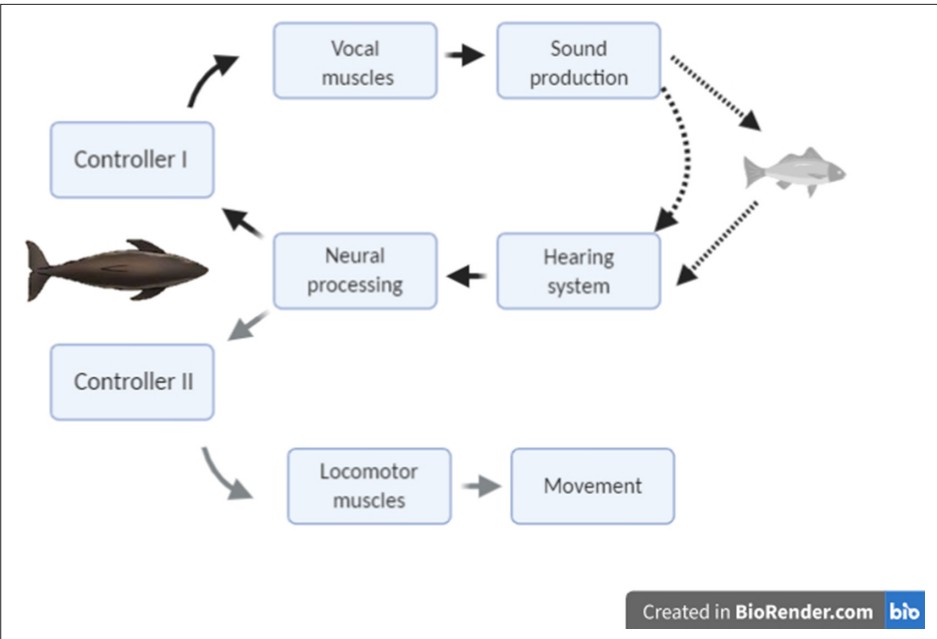

**Figure 1.** Conceptual schematic of the hypothesised feedback control mechanisms that act in response to changing target range in echolocation-mediated foraging. The dark lines indicate acoustic paths in echolocation. The grey line indicates movement of the predator towards the prey.

obtaining enhanced temporal resolution to track evasive prey at the expense of a short sensing range (*Wisniewska et al., 2016*). In analogy with engineered sonars, the working model for animal biosonar assumes that neural echo processing occurs on a click-by-click basis in which the next click is produced after echoes from the current click are detected and processed (*Au, 1993*). During buzzes, bats can click at up to about 160 /s (*Elemans et al., 2011*) requiring very short neural latencies to maintain such click-synchronised processing. However, in toothed whales, the click rates attained during buzzes (>500/s in porpoises, *Wisniewska et al., 2012*) would require neural and muscular responses two orders of magnitude faster than primate visual reflexes (*Kawano, 1999*) to keep up with the information flow. This raises the question of whether toothed whales have an extreme processing capability for acoustic information or if they instead integrate echo information over successive clicks (*Kothari et al., 2018*; *Ladegaard et al., 2019*) giving a processing time more similar to vision. In the latter case, why produce so many clicks during close approaches?

The modulation of clicking rate in echolocation to control information flow serves a similar function to short-latency eye movements in vision, leading us to hypothesise that echo information must guide two inter-related control loops during hunting (*Figure 1*): a kinematic response loop controls the heading, posture, and locomotor rate so as to intercept prey, while a sensor-motor response loop maintains attention fixed on the moving target by continual adjustment of the clicking rate. On that basis, we predict that sudden prey movements during close approaches should provoke tightly coupled mechanical and sensor-motor responses in echolocating animals similar to those shown by visual predators. However, despite echolocation being the main sensory mode for hunting in one of four species of mammals (*Madsen and Surlykke, 2013*), the sensory feedback that echolocating predators receive from movements of their prey has received little attention. *Wisniewska et al., 2016* reported clicking rate dynamics of wild harbour porpoises during buzzes that appeared to be associated with prey movements but offered no analysis. The only reported study of predator responses during echolocation buzzes is for wild bats approaching suspended prey that were moved by hand. Rather than adjust clicking rates, these bats aborted buzzes some 100 ms after strong target movements (*Geberl et al., 2015*), perhaps indicative of a neural processing delay, but the decision to end a buzz may well result from different neural processing than that involved in prey tracking.

The scant literature on sensory feedback in echolocation owes much to the inherent difficulty of measuring simultaneously the motor responses of predators and movements of prey at high time

resolution. This has seldom been achieved outside of instrumented enclosures where movements can be tracked with high-speed cameras, but where ecologically relevant stimuli are hard to emulate. Echolocating toothed whales provide an excellent model system in which to measure the coupled kinetics of free-ranging predators and prey. Sound and movement logging tags (DTAGs; *Johnson and Tyack, 2003*) record the outgoing echolocation clicks of toothed whale species and, for some species, also detect returning echoes from prey (*Johnson et al., 2004*; *Wisniewska et al., 2016*). They simultaneously record the fine-scale movements of the tagged animal, allowing quantification of both the sensory and locomotor responses of predators to movements of their prey. Crucially, these tags sample the sensory scene at exactly the same rate as it is acquired by the animal, that is, the rate at which clicks are produced.

Here we used high-resolution DTAGs on two species of echolocating toothed whales living in very different habitats to study biosonar responses to prey movements in the wild. Specifically, we tested the hypotheses that prey movements trigger neuromotor feedback during buzzes, and that this feedback operates at the extreme speeds needed to keep pace with the high clicking rate in buzzes. We show that both species make stereotyped biosonar adjustments when prey attempt to escape during close approaches, but the apparent latency of these echo-kinetic responses is much longer than the inter-click intervals (ICIs) in buzzes. Given the evolutionary importance of such a feedback system, we further hypothesised that it would be stimulated by any rapidly moving target during a close approach, facilitating controlled studies of biosonar responses. To test this, we trained harbour porpoises to approach a moveable target while wearing a biologging tag, enabling direct measurement of biosonar-mediated sensory and kinematic response latencies as a function of target movement.

## Results

### Biosonar responses to prey movement in free-ranging animals

Echograms visualising the acoustic scene during prey capture buzzes in wild harbour porpoises (*Phocoena phocoena*, abbreviated to Pp) and Blainville's beaked whales (*Mesoplodon densirostris*, abbreviated to Md) frequently show evidence of evasive prey that launch sudden escape attempts as the predator approaches (*Figure 2A and B*, *Figure 2—figure supplements 1 and 2*). Prey that accelerate away from the predator can quickly move beyond the acoustic depth of field (i.e., the buzz ICI times one half of the sound speed) requiring a rapid increase in ICI by the predator to avoid ambiguous target ranging. The ICIs used in buzzes when targeting evasive prey show dynamics that seem to correspond to changes in prey range (*Figure 2A and B*, *Figure 2—figure supplements 1 and 2*), suggesting a tight sensor-motor feedback loop. To verify that these ICI changes are linked to prey escapes, we plotted the proportion of outward depth-of-field adjustments (i.e., increasing ICIs) in time bins synchronised with the first detectable prey movement in buzzes. Echograms with clear echo traces showing sudden prey movements suitable for this timing analysis are sparse. Pooling data from six harbour porpoises, we found 76 buzzes in which the prey escape speed exceeded the predator closing speed by more than 0.25 m/s, leading to clear V-shaped echo traces. These buzzes (*Figure 2C*) showed strong positive ICI adjustments that differ significantly from control buzzes (i.e., with randomly reassigned target movement times) beginning in the 50–100 ms time bin after the start of prey escapes. Suitable echograms for Blainville's beaked whales were less common. After relaxing the selection criteria to accept buzzes with any sudden detectable increase in prey speed away from the whale, we pooled 36 examples from seven individuals. The resulting ICI distributions (*Figure 2D*) showed strong positive adjustments that differed significantly from control intervals beginning 100–200 ms after initial prey movement. Thus, in both toothed whale species foraging in the wild, evasive movements by prey appear to be consistently matched by biosonar adjustments that maintain the prey within the unambiguous depth of field with a feedback loop latency of ~50–200 ms. ICIs during buzzes decrease to <2.5 ms for harbour porpoises and <3.5 ms for Blainville's beaked whales, meaning that at least 20 clicks are produced during this latency time.

Energetic prey targeted by harbour porpoises can make multiple escape attempts within a buzz providing an opportunity to examine ICI responses to repeated cues (*Figure 3A*). Plotting prey range against the unambiguous depth of field (equivalent to an input-output plot of a control system) revealed distinctive counterclockwise loops due to the ICI response latency (*Figure 3B*). To estimate

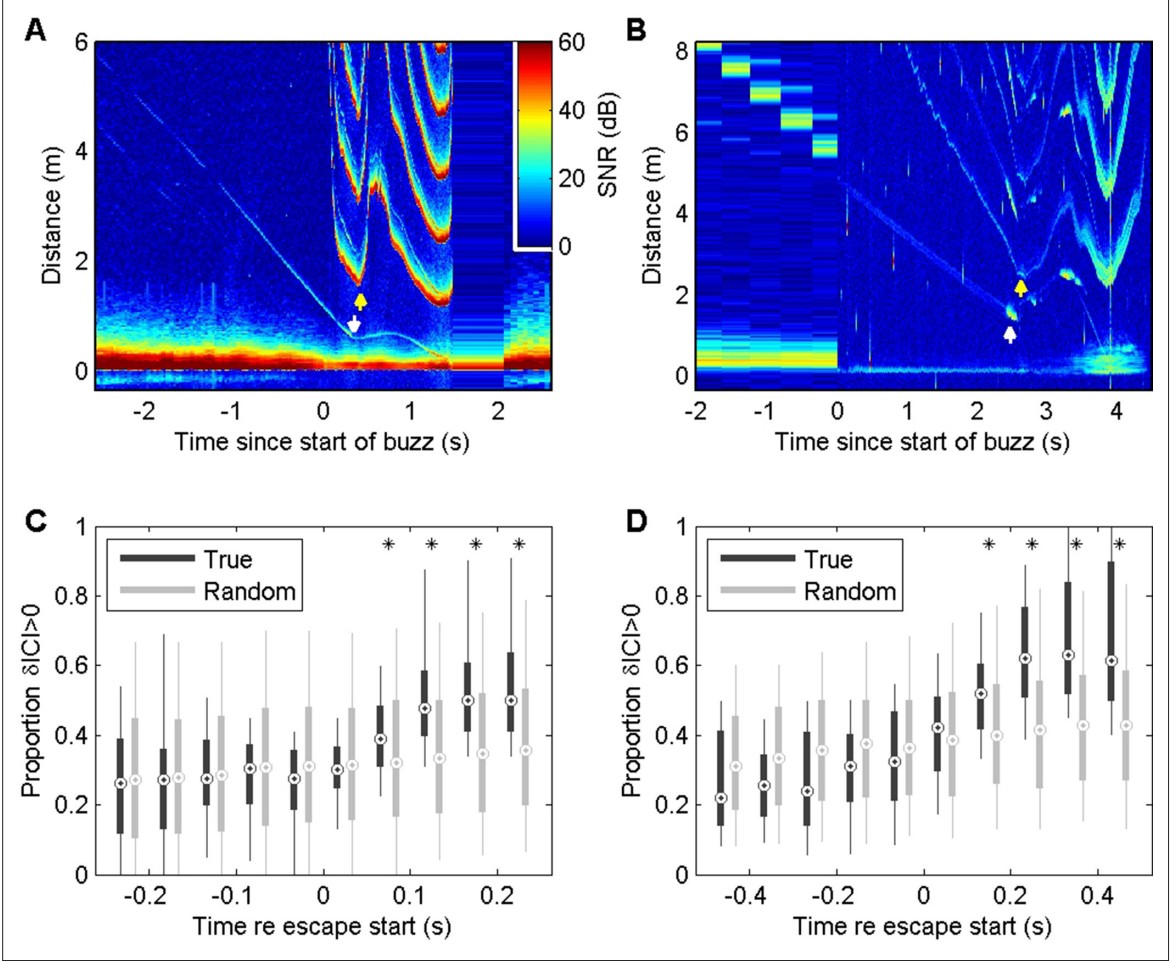

**Figure 2.** Echograms from a wild harbour porpoise (**A**) and a Blainville's beaked whale (**B**) during close prey approaches (buzzes) show prey escape attempts (white arrows) and dynamics in inter-click interval (ICI, yellow arrows) that appear to be causally related. These plots show echo strength (colour-coded by signal-to-noise ratio [SNR], in decibels [dB] as a function of distance from the predator [vertical axis] and time with respect to the start of the buzz [horizontal axis]). See *Figure 2—figure supplements 1 and 2* for a guide to interpreting these plots. Boxplots below show the proportion of positive changes in ICI ($\delta$ ICI, i.e., an expansion of the acoustic depth of field) as a function of time before/after the first prey response in 74 buzzes by six harbour porpoises (**C**) and 36 buzzes by seven Blainville's beaked whales (**D**). Due to differences in click repetition rate, 50 ms bins are used for harbour porpoise and 100 ms bins for Blainville's beaked whales. Light grey boxes show the changes in ICI for randomised control data. * indicates bins in which >95% of observed proportions exceeded the randomised proportion of positive $\delta$ ICI for control data.

The online version of this article includes the following figure supplement(s) for figure 2:

**Figure supplement 1.** Generating an echogram from on-animal sound recordings involves several steps.

**Figure supplement 2.** Individual coloured bars, generated following the process outlined in *Figure 2—figure supplement 1* are stacked together and rotated by 90° to produce an echogram (left).

this latency, we advanced the ICI time series in 5 ms steps until the areas within the loops were minimised. An advance of 90 ms collapsed most of the loops in the delay compensation plot (*Figure 3C*), suggesting that successive prey movements elicit responses with self-similar latency. In comparison, the magnitude of the ICI responses varied widely. Although ICI changes proportionate with prey escape movements occurred in some cases (*Figure 3A*), over-compensation was more typical (*Figure 2—figure supplements 1 and 2*), leading to depths of field that extended well beyond prey range when prey attempted to escape. However, echoes from schools of prey, or from reflectors such as the seafloor or sea surface, can have dynamics which greatly exceed the speed of single prey, requiring rapid outward ICI adjustments to avoid range ambiguity (*Figure 4*).

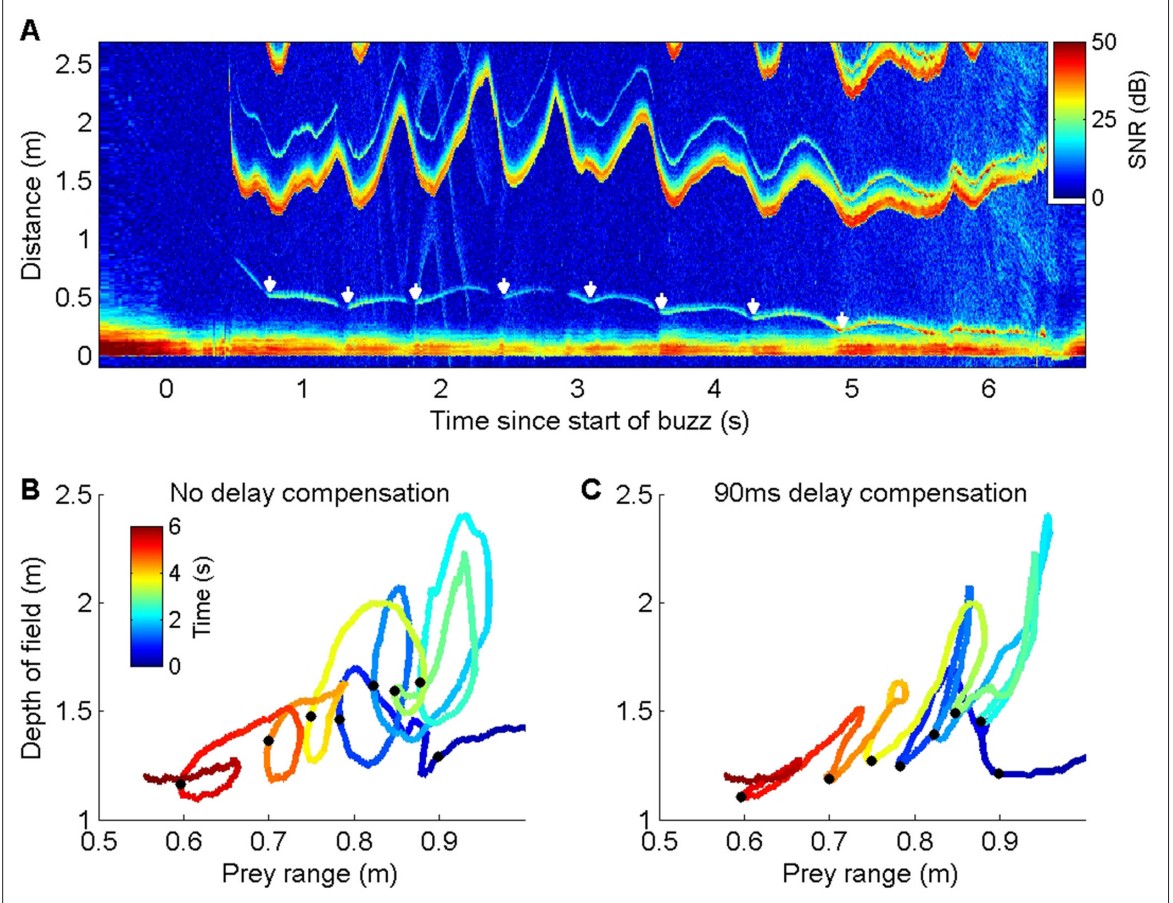

**Figure 3.** ICI responses of a wild harbour porpoise to repeated prey escape attempts. (**A**) Echogram of a buzz performed by a wild harbour porpoise while chasing an elusive prey. The prey makes a series of escape attempts, marked with white arrows, and the inter-click interval (ICI) (the first yellow-red line above 1 m distance, here converted by the echogram scaling to the acoustic depth of field in metres, i.e., ICI × sound speed/2) varies cyclically with each prey movement. (**B, C**) Plots of prey range versus acoustic depth of field for the same echogram with the times of prey escapes indicated by black dots. Anti-clockwise loops in (**B**) indicate that ICI lags behind prey movements. Advancing the ICI by 90 ms flattens the loops (**C**), indicating that the ICI is consistently delayed by about this amount with respect to prey movements throughout the buzz.

## Echolocation responses of trained animals to target movements

In wild predator-prey interactions, predators frequently strike at prey at about the same time that the prey responds, raising the possibility that ICI adjustments are timed based on predator strikes (i.e., implying an anticipatory or feed-forward control scheme) rather than based on prey movement. To exclude this potential confound, we designed an experiment in which captive harbour porpoise approached a target that could be moved suddenly. Echograms during target approaches (*Figure 5*) show that this experimental design successfully replicated the sharp speed changes of prey in wild porpoise chases and thus provide reliable cues for timing biosonar and kinematic responses.

Responses to fast target movement were evaluated in 43 and 31 trials for two captive porpoises, Freja and Sif, respectively. ICI responses, parameterised by the proportion of positive ICI changes, showed similar latency as for wild porpoises with response times of 50–100 ms for Freja and 100–150 ms for Sif (*Figure 6A and B*). Whole-body kinematic responses, inferred from the differential of on-animal acceleration measurements (jerk), were detectable with latencies of 0–50 ms for the two porpoises, although strong jerk responses were more clearly evident at 50–100 ms (*Figure 6C and D*).

As the target was pulled at varying speeds by hand, the trials could be ranked according to the magnitude of the initial target movement cue based on the signal recorded from the built-in accelerometer. Using root-mean-square (RMS) target acceleration as a proxy for this cue, we plotted ICI and RMS jerk as a function of cue magnitude (*Figure 7*). Whereas the analysis of *Figure 6* included all responses, here we set detection thresholds so as to focus on large biosonar and movement

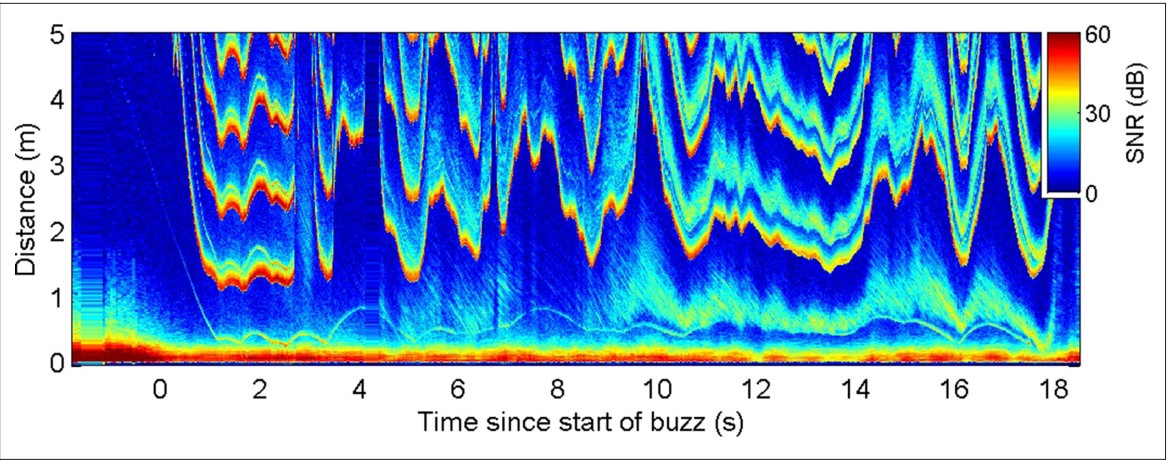

**Figure 4.** Echogram during a buzz recorded from a wild harbour porpoise hunting near the seafloor. Inter-click interval (ICI) adjustments (evidenced by the first yellow-red line in the plot at range >1 m) appear to respond to rapid changes in overall echo cross-section rather than the movements of the closest prey (e.g., at 9.5–10 s when the seafloor echo expands, due most likely to a change in orientation of the porpoise, while the prey trace stays at approximately the same range).

adjustments that might be broadly comparable to saccades in vision. The latency of these strong responses was consistently longer than the responses shown in *Figure 6* and tended to increase with decreasing target acceleration. Using a 5 ms threshold (corresponding to a 3.75 m depth of field) to detect strong outward ICI adjustments, the latency of these large-scale responses was inversely correlated with RMS target acceleration ($r^2 = 0.3$, $p<0.0001$). Kinematic response latencies, using a 300 m/s$^3$ threshold on RMS jerk to detect the onset of strong responses, were similarly inversely correlated with RMS target acceleration ($r^2 = 0.2$, $p<0.001$), albeit with more variability.

## Discussion

Toothed whales use echolocation in a deliberative mode to stalk unwary prey from long ranges, but must transition to a reactive mode when close enough that their bow-wave can be detected by prey (*Wisniewska et al., 2012*). In this discrete-time sensory system, there is an unavoidable trade-off between sensing range and temporal resolution (*Madsen and Surlykke, 2013*): to detect rapid prey movements, echolocators must sample at a high rate. However, under the prevailing click-by-click model for echolocation processing (*Au, 1993*), such fast sampling would require infeasible neural and muscular speeds, leaving uncertain how toothed whales respond to evasive prey. Our results from both wild animals and controlled trials demonstrate that toothed whales make sensor-motor responses to sudden target movements during echolocation buzzes but that the response latencies, although fast compared to other mammalian sensory systems, span multiple clicks. Our sample size (eight Md, and six wild and two captive Pp) is constrained by the difficulty of tagging wild toothed whales and by the availability of suitably trained captive animals. However, the consistent speed and stereotypy of responses in both natural foraging interactions and controlled trials strongly support an acute echo-mediated sensory feedback loop that is responsive to the evasive manoeuvres of prey. We propose the term echo-kinetic for these responses in analogy with the optokinetic responses that control ocular tracking in visual predators.

Our results show that high-sampling-rate buzzes, which are a defining characteristic of toothed whale echolocation (*Wisniewska et al., 2014*), delimit a reactive sensory mode in which the greatly increased temporal resolution allows detection of fast prey motion while rapid feedback control of click rate maintains unambiguous tracking of escaping prey. The universality of this sensory strategy in toothed whales is supported by the similarity of our results from two toothed whale families that diverged some 21 mya (*McGowen et al., 2009*) and occupy very different niches: Blainville's beaked whales dive to great depths to forage on fish and squid in and below the deep scattering layer (*Arranz et al., 2011*), whereas harbour porpoise forage epipelagically and benthically on small shallow-dwelling fish (*Wisniewska et al., 2016*). Moreover, similar biosonar responses were consistently elicited from

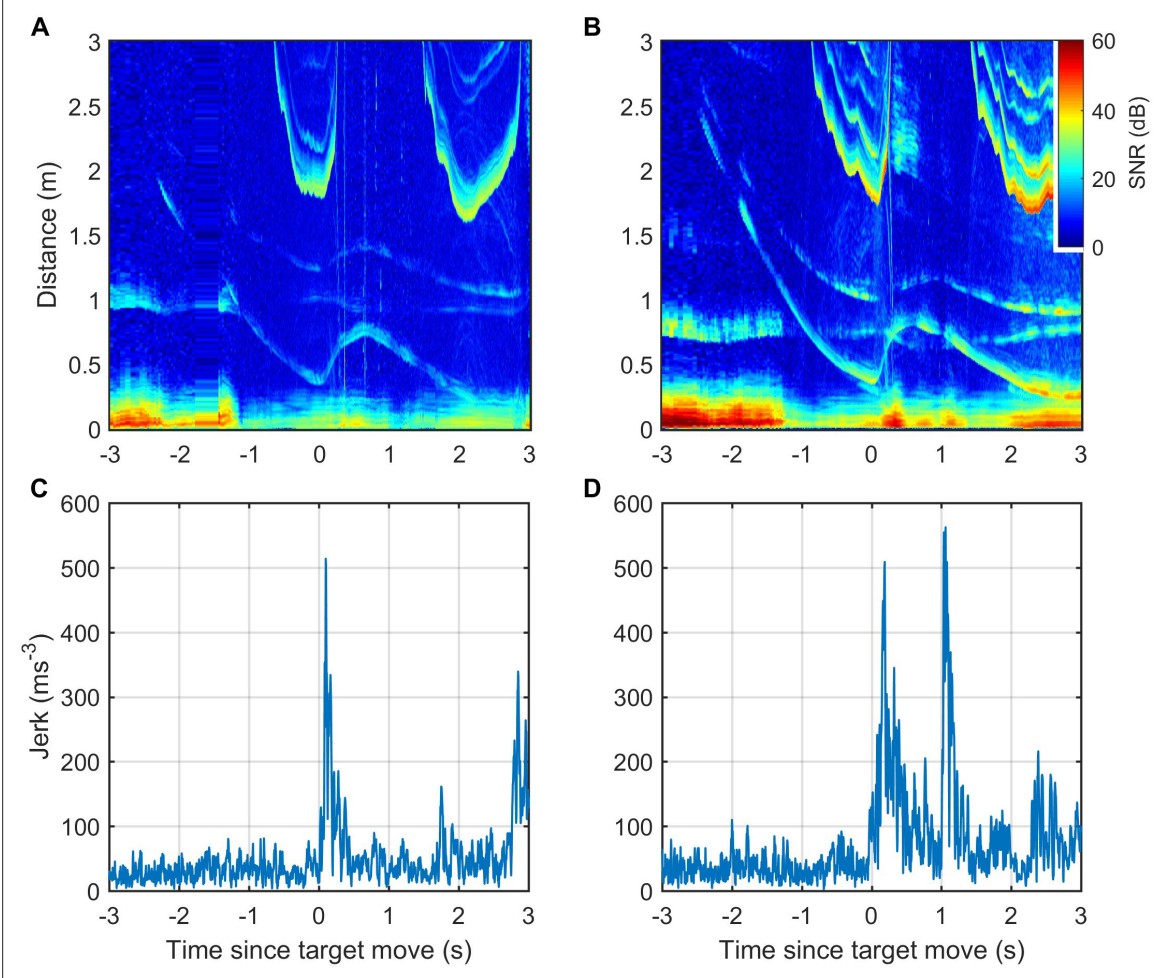

**Figure 5.** ICI and kinematic responses of two trained harbour porpoise approaching a moving target. (**A, B**) Echograms recorded by tags attached to captive harbour porpoises Freja (**A**) and Sif (**B**) during controlled target movement trials are similar to echograms recorded during elusive prey captures by wild porpoises. The target echo in the echograms is the trace that moves suddenly outwards just after time 0. Other echo traces are from the water surface and from the target and water surface together (i.e., the sound reflects from the target and then the water surface, or vice versa, before returning to the porpoise). (**C, D**) Jerk magnitude, that is, the mechanical response of the porpoises, recorded by the tags and synchronised with the echograms.

captive porpoises by sudden movements of an artificial target hinting that this reflex-like behaviour is deeply embedded in the neural substrate of toothed whales. This leads us to propose that acute sensor-motor feedback during buzzes is a fundamental feature of toothed whale echolocation that has enabled hunting of nutritionally-valuable muscular but reactive prey.

The measured response latencies ( *Figure 2* , *Figure 6*) show that tight sensor-motor feedback in echolocation buzzes can be achieved without extreme neural processing speeds. Echo-kinetic response latencies of 50–200 ms in our study are comparable to short-latency eye movements in primate vision (*Land, 1999*; *Kirchner and Thorpe, 2006*; *Erkelens, 2006*) and to vocal response latencies to passive acoustic cues in dolphins (*Ridgway, 2011*) and porpoises (*Wensveen et al., 2014*). However, these response latencies are more than 20× longer than the 2.5–3.5 ms ICI during buzzes, demonstrating that echolocating whales process and respond to echo information during prey approaches much more slowly than they acquire it. Put another way, the maximum information bandwidth (i.e., 1/(2 × ICI) Hz, by Nyquist theorem) is some 40× greater than the maximum control bandwidth that can be achieved given the response delay, that is, approximately 1/(4× response latency) Hz (*Åström, 1997*). This implies strongly that echo processing and control decisions during buzzes are decoupled from click rate rather than occurring on a click-synchronous basis as widely assumed (*Au, 1993*). This conjecture is consistent with the proposed processing mode of click packets produced during long-range echolocation of dolphins (*Ladegaard et al., 2019*; *Finneran, 2013*).

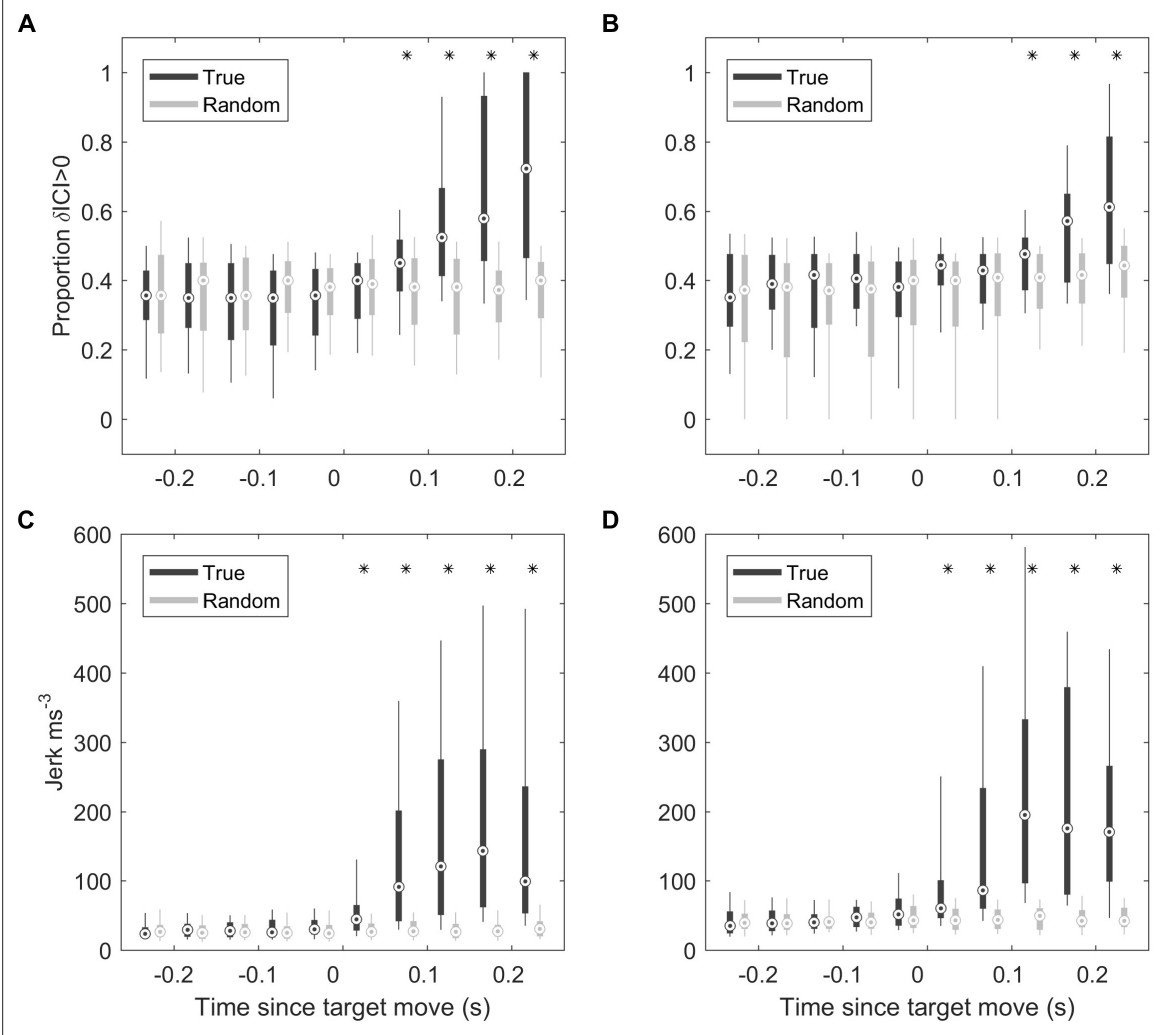

**Figure 6.** ICI and kinematic response latencies of two trained harbour porpoise during moving target trials. (**A, B**) Boxplots of proportion of positive δ ICI (i.e., outward adjustments in the acoustic depth of field) in 50 ms bins synchronised to the start of the target movement for Freja (**A**) and Sif (**B**). Grey boxes indicate the results for control buzzes in which the target movement time was randomised. * indicates bins in which >95% of observed proportions exceeded the randomised proportion of positive δ ICI for control data. ICI: inter-click interval.(**C, D**) Boxplots of root-mean-square (RMS) jerk in 50 ms bins synchronised to the start of target movement for Freja (**C**) and Sif (**D**). Grey boxes indicate the results for control jerks in which the target movement time was randomised. * indicates bins in which >95% of observed jerks exceeded the randomised jerks.

The dramatic bandwidth mismatch between information gathering and feedback control in echo-location buzzes begs the questions of how these processes are decoupled and what purpose this serves. Unlike bats, which generate calls individually by contractions of super-fast vocal muscles (*Elemans et al., 2011*), we propose that the extreme click rates in toothed whale buzzes are achieved by a free-running pneumatic oscillator comprising the pressurised pre-narial air space and phonic lips (*Au and Suthers, 2014*). This oscillator produces buzz clicks at a rate determined by the air pressure and the tension of the phonic lips, both of which can likely be controlled asynchronously with respect to click production. In this model, control decisions, formed after observing echoes from tens of clicks during buzzing, modulate the rate of clicking with the objective of maximising temporal resolution subject to the constraint that the ICI is consistently greater than the two-way travel time to the target.

The high clicking rate in buzzes enables rapid detection of prey responses but may provide other benefits, when combined with appropriate feedback mechanisms, such as (i) signal-to-noise ratio improvement of weak echoes by integration over multiple clicks (i.e., using integral control); (ii) speed-based processing of echoic scenes to predict target motion (via differential control); and (iii) detection of modulations in echo level (e.g., due to prey tail-beats, *Wisniewska et al., 2016*) that may be the

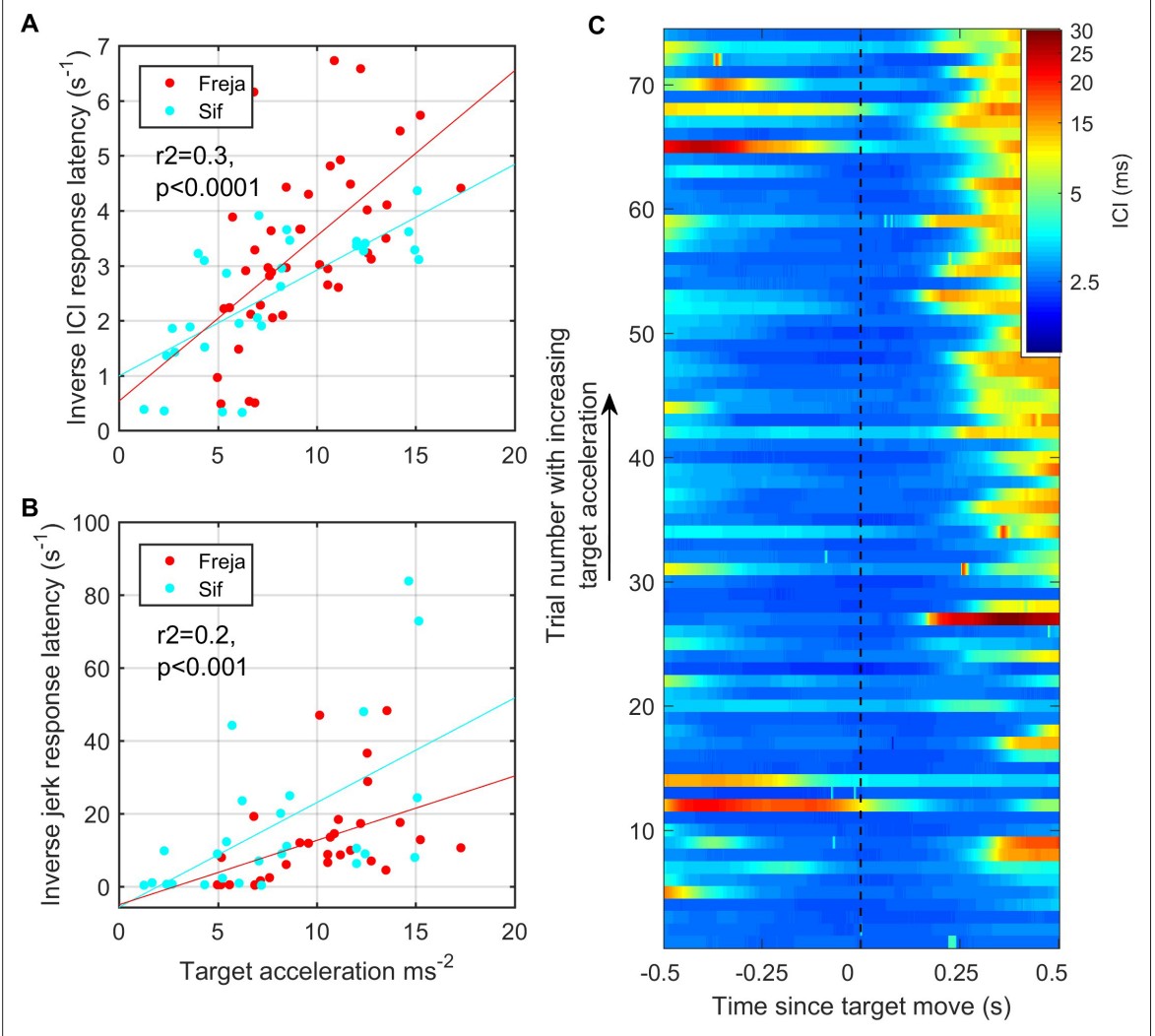

**Figure 7.** Fast target movements were associated with faster inter-click interval (ICI) and kinematic responses (**A, B**). (**A**) shows the time delay between the start of target movement and the ICI exceeding 5 ms (corresponding to a depth of field greater than 3.75 m) for each trial for the two porpoises as a function of the root-mean-square (RMS) target acceleration. (**B**) shows the time delay from the start of target movement to a jerk exceeding 300 ms$^{-3}$. The inverse latency is plotted in both cases to linearise the relationships for regression analysis. Reported regression results are for pooled data from Freja and Sif. (**C**) Stacked plot of ICI synchronised to the start of target movement for all analysed trials by Freja and Sif. Each horizontal line represents a single trial, and the colour indicates the ICI on a log scale. Trials are ordered in terms of increasing target acceleration such that trials at the top of the plot have the fastest initial target movement. ICI responses occur with shorter latency in these fast trials as compared to the trials with slower target movement at the bottom of the plot.

earliest cues of prey responses, while avoiding aliasing in this discrete-time sensor. The high click rate in buzzes effectively forms a temporal fovea, akin to the spatial fovea in many visual predators, matched to the burst movement rates of the relatively small prey targeted by most toothed whales. This ensures the observability of prey behaviour and enables control tactics that counter unpredictable prey movement.

Echograms recorded during wild encounters with evasive prey (*Figures 2–4*) hint at two control tactics that may be employed. When the change in target range is dominated by predator movements, a gradual upward adjustment of clicking rate is sufficient to track the changing spatial relationship of predator and prey. In comparison, unexpected rapid prey movements often provoke large adjustments in the biosonar rate in which the acoustic depth of field is rapidly expanded at the expense of temporal resolution. This suggests a layered control with smooth tracking during stalking and when prey move predictably, but occasional saccade-like ballistic increases in clicking rate during chases. Such layered control actions may also accommodate the dynamics of schooling prey, which

can quickly switch between cohesion when being pursued and dispersion when escaping (*Couzin and Krause, 2003*). Similar to vision (*Erkelens, 2006*), the saccadic biosonar adjustments occurred with longer latencies in captive trials, compared to average responses, suggesting that large adjustments may be employed when targets move away sufficiently rapidly for there to be a risk of ambiguous echo ranging, that is, the echo delay exceeding the time between outgoing clicks.

While information bandwidths in echolocation are likely linked to prey dynamics, the control bandwidths (i.e., the speed with which the system can respond to changes) may be more matched to the size and manoeuvrability of the predator given that size influences both the rotational inertia of the body and the length, and therefore contraction rate, of muscles (*Domenici, 2001*). We hypothesised the existence of two control loops in echolocation-guided hunting, controlling, respectively, the acoustic depth of field and the swimming kinematics (*Figure 1*). We have been able to demonstrate the biosonar feedback loop in both wild and controlled settings, but full-body kinematic responses to prey movement are confounded in wild predator-prey interactions by the predator's own striking actions. However, our controlled studies demonstrate that porpoises make an accelerative response to target movement with latency roughly comparable to the biosonar response and it seems very likely that wild animals would have similar kinematic responses. Therefore, our finding of longer biosonar response latencies in Blainville's beaked whales, which are three times the size of harbour porpoises, suggests that control bandwidths may scale inversely with predator size. In effect, selection pressure on higher control bandwidths may be opposed by the increasing energetic cost of fast movements in large animals. An additional constraint on biosonar control bandwidth arises in the largest toothed whale species, sperm whales, due to the separation of the brain and sound source (located at the anterior tip of the nose) which may be more than 3 m apart. Even with highly myelinated nerves, the conduction delay (perhaps 30 ms) of neural signals to the sound source will be comparable to, or exceed, the time between successive clicks in buzzes (e.g., 10–20 ms, *Fais et al., 2016*). The decoupling between information flow and feedback control proposed here, in concert with a self-running pneumatic oscillator at the sound source, may have been instrumental in permitting such extreme cranial telescoping.

Thus, despite the overt differences between echolocation and vision, the response bandwidths and layered control inferred here for toothed whale echolocation are remarkably similar to those in primate vision (*Kirchner and Thorpe, 2006*), with response times of the order of 0.1 s, likely limited in both auditory and visual senses by higher-order processing and muscle contraction speeds. The apparent universal use of buzzes during capture of moving prey by echolocators (*Madsen and Surlykke, 2013*) suggests that extreme sensory sampling rates, guiding fast echo-kinetic responses, may have been a critical development, parallel to optokinetic responses in visual predators, enabling echolocation to be used to hunt agile prey, as opposed to just navigation and prey search. Our results, therefore, reveal strongly convergent neural sensor-motor feedback loops between vision and echolocation that are key for sensing dynamic spatial relationships with small prey. The non-invasive experimental approach developed here enables measurement of neuro-sensory dynamics while animals solve vital real-world problems, opening the way for a deeper understanding of ecological drivers on sensor performance in the wild.

## Materials and methods
### Echolocation in free-ranging animals
Sound and movement recording DTAGs were attached with suction cups to the anterio-dorsal surface of wild harbour porpoises (Pp, n = 6) in inner Danish waters between 2012 and 2018, and Blainville's beaked whales (Md, n = 8) off El Hierro in the Canary Islands between 2004 and 2013. DTAGs (v3 and v4) were attached to harbour porpoises bycaught in pound nets as they were removed from nets (for details, see *Wisniewska et al., 2016*). For beaked whales, DTAGs (v2 and v3) were delivered to free-swimming animals using a hand pole from a small inflatable boat (for details, see *Aguilar de Soto et al., 2012*). The tags sampled sound from mono or stereo hydrophones with 16-bit resolution and a sampling rate of 500 or 576 kHz (Pp), and 192 or 240 kHz (Md) (clipping levels of 170–175 dB re 1 µPa). Tags also sampled depth sensors, and tri-axial accelerometers and magnetometers, at sampling rates of 200–625 Hz/channel (Pp) and 50 Hz/channel (Md). Tags automatically released from animals after a pre-programmed interval and were recovered by VHF radio tracking.

Data processing was performed in Matlab 2016a (MathWorks Inc). Spectrograms of the on-animal sound recordings were examined to identify rapid click sequences (buzzes) during foraging. For each buzz, the production times of clicks were determined using a supervised click detector with approximately 50 μs accuracy. Clicks from the tagged animal were distinguished from those of other nearby animals by the consistent angle-of-arrival (on stereo tags) and broader frequency range of the former. Buzzes were defined as intervals in which the ICI was below 0.013 s (Pp) (*Wisniewska et al., 2016*) or 0.1 s (Md) (*Johnson et al., 2006*) for at least 0.5 s. Echograms were formed for each buzz by first bandpass filtering the sound (Pp: 100–250 kHz; Md: 25–60 kHz) and then computing the amplitude envelope using the Hilbert transform. Segments of envelope synchronised to each click were extracted and displayed as coloured bars with width equal to the click's ICI, resulting in an echogram display with axes of time and distance (*Johnson, 2014*). Body movement during buzzes was quantified from the norm of the jerk, that is, the vector magnitude of the change rate of the tri-axial acceleration signals (*Johnson et al., 2004*).

Buzz echograms were selected for timing analysis based on visual inspection. Echograms with unclear prey echo traces or with substantial interference (e.g., echoes from the sea surface, seafloor, or other organisms) were rejected. The remaining echograms were examined for indications of prey escape attempts. These appear as sudden changes in the slope of prey echo traces (*Figure 2—figure supplement 1* and *Figure 2—figure supplement 2*) reflecting a step change in the closing speed between predator and prey as the prey accelerates away (*Wisniewska et al., 2016*). As prey reactions are typically fast, the onset time of the slope change in the echo trace is usually well-defined. The first such reaction time in each buzz echogram was selected manually with ~10 ms accuracy, and traces with unclear or gradual slope changes were rejected. Potential biosonar adjustments to these prey movements were quantified by the proportion of positive changes in ICI (suggesting an outward adjustment of the depth of field) in 50 ms (Pp) or 100 ms (Md) bins, spanning from 500 ms before to 500 ms after each prey response time. As ICI varies continuously throughout buzzes, these bin sizes were chosen as a compromise between temporal resolution and rejection of noise from routine ICI variations. The wider bin size for Md reflects the longer ICIs produced during buzzes by these larger animals.

To determine the probability of chance associations between target movement and ICI changes, a bootstrap method was applied for each species. The same biosonar response metric was computed 1000 times for randomly selected pairs of buzzes in which the prey movement time from one buzz was applied to another buzz. Specifically, the time elapsed between the start of the buzz and the onset of the prey response in the first buzz of each pair was added to the start of buzz time in the second buzz to give a mock prey move time from which to reference the analysis time bins. A significant deviation from chance was concluded for each time bin in which >95% of the observed proportions exceeded the randomised proportions.

## Echolocation responses of trained animals to target movements

Experiments were carried out in an 8 × 12 m semi-natural facility at Fjord & Bælt, Kerteminde, Denmark, in May 2017 with two harbour porpoises (Sif and Freja, both female). At the time of the experiments, Sif was 1.6 m in length, 14 years old, and had been housed at Fjord & Bælt since 2004. Freja was 1.58 m in length, 20 years old, and had been held at the facility since 1997. Both porpoises were trained to locate and intercept a 50.8-mm-diameter aluminium sphere while wearing opaque soft silicone eye cups. The target sphere contained an embedded hydrophone (flat [±2 dB] frequency response from 1 to 160 kHz) and two-axis accelerometer (flat [+0/–3 dB] frequency response from 0 to 2 kHz, axes oriented horizontally), and was suspended in the water via a 0.8 mm nylon string to a depth of approximately 1.5 m. A 1.2-mm-diameter screened cable carrying the accelerometer and hydrophone signals from the target was loosely attached to the nylon string and connected out of the water to a three-channel synchronous 16-bit analog-to-digital convertor sampling at 500 kHz (National Instruments, Austin, TX). A second nylon line running horizontally from the sphere to the side of the pool was used to move the target during trials. Animals were equipped with a DTAG v4 sound and movement recording tag attached 5 cm behind the blow-hole with silicone suction cups. This tag contains a single hydrophone sampled at 576 kHz (flat [±2 dB] frequency response from 1 to 150 kHz, clipping level of 175 dB re 1 μPa) together with a tri-axial accelerometer sampled synchronously at 200 Hz and a tri-axial magnetometer and pressure sensor sampled at 50 Hz.

For each session, one of the two porpoises was introduced into the pool and stationed approximately 8 m from the target until given a cue to perform the target interception task. If the animal intercepted the target by touching it, it was bridged with a whistle and received a fish reward upon returning to station. In randomly selected trials, the target was moved manually approximately 30 cm by pulling vigorously on the horizontal line when the porpoise approached within one body length. The line was held slack prior to this to limit any early anticipatory target movement. Target movement was selected pseudo-randomly for each trial between fast, slow, and no movement, with a maximum run of two equal conditions. Up to 20 trials were performed with each animal per day for a total of 150 trials over 4 days for the two animals.

Echolocation clicks were detected in the animal-attached DTAG recordings using a supervised detector. The time offset between the tag and the National Instruments recordings was then determined for each trial by matching click sequences between the tag and the target hydrophones (max. timing error due to acoustic propagation of ~1 ms). Echograms were then assembled from the tag data as described above. Sudden changes in the closing speed between the target and the porpoise due to rapid movement of the target generated the same distinctive slope changes in echo traces as observed in wild predator-prey interactions. To maximise timing accuracy, trials were only selected for analysis if the target attained a speed greater than the porpoise's approach speed (approximately 1 m/s), resulting in a V-shaped echo trace. The apex of the V was then taken as the reference time for calculating response latencies. The target acceleration (as measured by the embedded accelerometer) began ~100 ms before this due to tightening of the line and rotation of the spherical target to align the tie point with the pull direction. Given the thin line and spherical target, neither of these movements generated significant echo signatures and the porpoise was therefore unlikely to detect the target motion until it is underway. We accordingly view the apex of the V in the echogram as a close indicator of the time at which the porpoise was first able to detect the target movement. As the target movement varied in each trial, the root-mean-squared target acceleration was calculated as a proxy for target motion. This was computed from the accelerometer embedded in the target by removing the fixed gravity component from each axis and then summing the squared signals from both axes over the 500 ms following the first acceleration transient.

Biosonar responses to target movement were quantified as for the wild toothed whales using the proportion of positive ICI changes in 50 ms intervals. Locomotor responses to target movement were assessed from the norm-jerk (*Ydesen et al., 2014*) calculated from the DTAG accelerometer data sampled at 200 Hz. To assess the probability of chance associations between target movement and ICI changes, the same bootstrap method was applied as for wild harbour porpoise and beaked whales (see above), that is, randomly selecting 1000 pairs of trials and applying the elapsed time between buzz start and target movement from one trial to the buzz of another trial.

## Acknowledgements

The captive harbour porpoises were cared for by Fjord & Bælt, Denmark, under permit nos. SN 343/FY-0014 and 1996-3446-0021 from the Danish Forest and Nature Agency. The wild harbour porpoise tagging study was funded by the Federal Agency for Nature Conservation under the contract Z1.2-5330/2010/14 and the BfN-Cluster 7 'Effects of underwater noise on marine vertebrates'. We thank J Kristensen, J Larsson, F Johansson, M Kjølby, and all the team at Fjord & Bælt for their expertise and support in carrying out the experimental element of the study with harbour porpoises. We thank K Beedholm of the Department of Bioscience, Aarhus University, for assisting with the recording setup. Thanks to members of the section of Marine Mammal Research and the Bioacoustics Lab at Aarhus University with particular thanks to S Elmegaard, C Malinka, L Rojano-Doñate, J Teilmann, S Sveegaard, J Balle, E Iglesias, L Kyhn, L Havmøller, P Tønnesen, K Sprogis, J Tougaard, A Galatius, L Mikkelsen, MV Jensen, L Hermannsen, P Meyer Soerensen, M de Freitas, M Dyndo, B McDonald, R Dietz, A Hansen, and B Hansen. Thanks also to the ULL team for their support for tagging beaked whales at El Hierro, especially to P Arranz, A Schiavi, C Reyes, C Yzoard, and J Marrero.

# Additional information

## Funding

| Funder | Grant reference number | Author |
|---|---|---|
| Bundesamt für Naturschutz | Z1.2 5330/2010/14 | Peter T Madsen |
| Horizon 2020 | 754513 | Mark Johnson |
| Aarhus University Research Foundation | | Mark Johnson |
| Danmarks Frie Forskningsfond | 6108-00355B | Peter T Madsen |

The funders had no role in study design, data collection and interpretation, or the decision to submit the work for publication.

## Author contributions

Heather Vance, Data curation, Formal analysis, Investigation, Validation, Visualization, Writing – original draft, Writing – review and editing; Peter T Madsen, Conceptualization, Methodology, Resources, Supervision, Writing – review and editing; Natacha Aguilar de Soto, Danuta Maria Wisniewska, Conceptualization, Writing – review and editing; Michael Ladegaard, Methodology, Writing – review and editing; Sascha Hooker, Supervision, Writing – review and editing; Mark Johnson, Conceptualization, Data curation, Formal analysis, Funding acquisition, Investigation, Methodology, Project administration, Software, Supervision, Validation, Visualization, Writing – original draft, Writing – review and editing

## Author ORCIDs

Heather Vance (iD) http://orcid.org/0000-0003-1027-3128
Peter T Madsen (iD) http://orcid.org/0000-0002-5208-5259
Natacha Aguilar de Soto (iD) http://orcid.org/0000-0001-9818-3527
Michael Ladegaard (iD) http://orcid.org/0000-0001-7559-9271
Mark Johnson (iD) http://orcid.org/0000-0001-8424-3197

## Ethics

The captive harbour porpoises were cared for by Fjord & Bælt, Denmark, under permits no. SN433 343/FY 0014 and 1996 3446 0021 from the Danish Forest and Nature Agency. Their care and all experiments are in strict accordance with the recommendations of the Danish Ministry of Food, Agriculture and Fisheries (issuing the permit to keep the animals), the Danish Ministry of the Environment (permit for catching the animals) and the Danish Council for Experiments on Animals (always contacted for permits when appropriate-but in the case of this study such permit was not required). Wild harbour porpoise tagging was carried out under permission from the Danish Forest and Nature Agency (NST-3446-00016) and the Animal Welfare Division (Ministry of Justice, 2010-561- 1801). Tagging of Blainville's beaked whales was conducted under a permit from the government of the Canary Islands. Field protocols were approved by the WHOI Institutional Animal Care and Use Committee.

## Decision letter and Author response

Decision letter https://doi.org/10.7554/eLife.68825.sa1
Author response https://doi.org/10.7554/eLife.68825.sa2

# Additional files

## Supplementary files

• Transparent reporting form

## Data availability

We have prepared a data archive and uploaded this to Dryad. The complete source data for this paper comprises several TB, the great majority of which are not needed to reproduce our findings. We have therefore extracted a subset that is sufficient to re-create echograms of the type seen in Figure

2A–B, Figure 3A, Figure 4, Figure 5A–B, and hence the results summary plots. Included for each trial is the acoustic data together with the timing of the prey/target movement and, for trials with trained animals, the speed of the target movement. Data comprises a single netcdf file for each analysed trial, and covers the 112 captive trials (controls and treatment conditions) as well as each analysed prey capture attempt from the wild data. The netcdf format allows us to attach metadata directly to each trial in a human-readable form. Further we have provided Matlab/Octave scripts to produce echograms from these data. The supplementary material included in the manuscript provides sufficient information to interpret these echograms.

The following dataset was generated:

| Author(s) | Year | Dataset title | Dataset URL | Database and Identifier |
|---|---|---|---|---|
| Vance H, Madsen P, Aguilar de Soto N, Wisniewska D, Ladegaard M, Hooker S, Johnson M | 2021 | Echolocating toothed whales use ultra-fast echo-kinetic responses to track evasive prey | https://doi.org/10.5061/dryad.n8pk0p2w1 | Dryad Digital Repository, 10.5061/dryad.n8pk0p2w1 |

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
