## [Decision Letter]

**Acceptance summary:**

This paper on echolocation-mediated responses to prey movements will be of interest to a broad audience, including ethologists and neuroscientists as well as those more generally interested in the natural world. Its strengths come from the use of data from both wild and captive animals of different species of toothed whales, as well as trained harbour porpoises, enabling generalization of the findings and conclusions on sensory-motor feedback. The authors show that these animals use their biosonar responses to follow sudden prey movements with speeds that resemble those of visual tracking responses in other species.

**Decision letter after peer review:**

Thank you for submitting your article "Echolocating whales use ultra-fast echo-kinetic responses to track evasive prey" for consideration by *eLife*. Your article has been reviewed by 2 peer reviewers, and the evaluation has been overseen by Miriam Spering as a Reviewing Editor and Andrew King as the Senior Editor. The following individuals involved in review of your submission have agreed to reveal their identity: Alexander J Werth (Reviewer #2).

Essential revisions:

Individual reviews below provide a list of issues that should be addressed for clarification purposes. Among them, a few stand out as more essential:

1. Consider changing the title to "echolocating whales and porpoises".

2. Emphasize / discuss sample size as a limitation.

3. A comment from the reviewing editor (who read this paper with interest, from the perspective of an oculomotor systems researcher): some extra care with regard to eye movement terminology would be appreciated. For example, the term "optokinetic reflex" is sometimes used in this paper to describe short-latency eye movements in general, for example, the papers referenced in l. 48 following are all about express saccades. Could the authors perhaps use the term short-latency eye movement? Other terms that have been used: orienting saccade, reflexive saccade.

*Reviewer #1 (Recommendations for the authors):*

Congratulations on a lovely manuscript. I have no major issues with it. I specifically liked the clever analyses presented in Figure 3 B-C for the wild animals. Though beyond the scope of this manuscript, it would be interesting to repeat the experiment with the captive animals tracking a target that moves repeatedly (not isolated trials). In this same line, future studies including physiological methods (ABRs and EMGs to track muscle responses) could provide further insight into the processing times for the pathway proposed by the authors.

It is my opinion that a more thorough discussion on prey tracking behaviors and the evolutionary arms-race would situate the findings in a broader context and improve the manuscript.

Some other comments:

Line 52: why do the authors suggest echolocation enables stealthy approaches? It is well documented that a number of prey can hear echolocating animals approaching. This sentence is confusing as it seems to imply echolocation provides more stealth than vision. Please reword.

Line 73: I suggest you add the citation Salles Diebold and Moss 2020, which also shows an increase in vocalization rate in bats tracking a moving prey item trajectory.

Line 83: please correct the sentence to "one in four species of mammals"

Line 212: please clarify that the acceleration of the target moved by hand was calculated with an accelerometer (as described in methods). As it reads it is unclear how controlled the target movements were.

Line 252: unit typo should read "m.s-3 " or "m/s3 " as in Line 218; not "ms-3".

Figure 7: From the figure it looks like the limit to how fast ICI change can occur after prey movement, ~200 ms for captive animals? While in the wild the authors show evidence for 100-200 ms delay after initial prey movement (line 141). Is there a limit to how fast the target is moved by hand that cannot replicate the shorter latencies seen in the wild?

*Reviewer #2 (Recommendations for the authors):*

1. Some readers might be confused with the word "whales" in the title, given that this study is based on porpoises too. Perhaps consider changing to "echolocating whales and porpoises"? (or cetaceans?)

2. In line 53, the authors present the entirely logical and reasonable argument that "prey are likely to detect the predator only when it is close," but because this follows a sentence about vision, it presents the idea-intentionally or unintentionally-that aquatic prey will detect their predators visually. In fact, it seems reasonable that most aquatic prey will use mechanoreception, particularly if a predator approaches from behind swimming prey, and if the prey are fish with sensitive lateral line systems. Even if the beaked whales are preying on squid (which typically rely on visual sensation), this mostly occurs in deep, dark waters where vision is of limited use.

3. Is it concerning that the two captive porpoises had similar but really non-overlapping response times ("50-100 ms for Freja and 100-150 ms for Sif"; lines 202-203)?

4. Can you provide some additional data about the two captive harbour porpoises (apart from the fact that both Sif and Freja are female; line 383). Size? Age? Length of time in captivity?

5. The analogy of a temporal fovea similar to the spatial (retinal) fovea is really nice, and quite evocative.

6. The sources supplied by the authors are good and up to date. In line 238, a source should be supplied for the statement about "the prevailing click-by-click model for echolocation processing". Who proposed, stated, or described/explained this model? Would it be Au 1993?

7. Two related thoughts: First, I have often heard, informally, that cetacean brains are very large but that, unlike most mammals, cetacean brains are largely devoted to processing auditory information. I am not a brain expert by any means, and although I doubt the veracity of this claim, I wonder if the authors might include here, perhaps in the Introduction of Discussion, a brief mention about cetacean brains and their relation to auditory processing abilities.

8. Second, I wonder about subconscious vs. conscious neural processing. I realize that this is not entirely pertinent to this study, but you've got me thinking about it. At the standard frame rate of television and movies (24 fps) we cannot (consciously) distinguish a series of discrete, separate still images-we see them, obviously, as a continuous stream of video input (even if we subconsciously detect subliminal cues in individual frames, for example). Might this relate in any way to this study? I like your two-part model of feedback control (Figure 1) of vocal muscles versus locomotor muscles. Could odontocete brains possibly recognize individual clicks, in the same way that we can "see" individual video frames, even if on another level they show responses that are decoupled from the click rate?

9. The color figures are crisp and easy to follow, and the captions make sense.

10. This manuscript is really very well written (I commend, and thank, the authors for the fine care they have taken in writing this), but an "Oxford comma" is needed in line 44 before the word "or" (third item in a list) to prevent possible confusion.

---

## [Author Response]

Essential revisions:Individual reviews below provide a list of issues that should be addressed for clarification purposes. Among them, a few stand out as more essential:1. Consider changing the title to "echolocating whales and porpoises".

We agree that 'echolocating whales' may not connote smaller delphinids to some readers. Our results suggest strongly that echo-kinetic responses are a common feature across the suborder odontoceti. We therefore propose to use the more precise term 'echolocating toothed whales'.

2. Emphasize / discuss sample size as a limitation.

Thank you. We note in our reply to Reviewer #2 that our sample size of wild porpoises and beaked whales (8 and 6, respectively), is considerably larger than stated by the reviewer (i.e., 2 of each species). However, the captive trials in the study were limited to 2 trained porpoises, and we have added text in the discussion to emphasise and discuss this limitation.

3. A comment from the reviewing editor (who read this paper with interest, from the perspective of an oculomotor systems researcher): some extra care with regard to eye movement terminology would be appreciated. For example, the term "optokinetic reflex" is sometimes used in this paper to describe short-latency eye movements in general, for example, the papers referenced in l. 48 following are all about express saccades. Could the authors perhaps use the term short-latency eye movement? Other terms that have been used: orienting saccade, reflexive saccade.

Thank you. We intend with the term 'optokinetic reflex' to refer to both smooth pursuit and saccadic responses that control eye position and orientation in target tracking. The suggested replacement term (short-latency eye movements) does not seem to us to have the same connotation of a feedback process. We have therefore opted to use both terms according to whether it is specifically the movement or the responsivity which is under discussion. We have also replaced one of the references at line 48 to include a citation covering both pursuit and saccadic latencies.

Reviewer #1 (Recommendations for the authors):Congratulations on a lovely manuscript. I have no major issues with it. I specifically liked the clever analyses presented in Figure 3 B-C for the wild animals. Though beyond the scope of this manuscript, it would be interesting to repeat the experiment with the captive animals tracking a target that moves repeatedly (not isolated trials). In this same line, future studies including physiological methods (ABRs and EMGs to track muscle responses) could provide further insight into the processing times for the pathway proposed by the authors.

Thank you for the kind words. Yes, this study has brought to our minds many avenues for future investigation and the reviewer's suggestions are most welcome.

It is my opinion that a more thorough discussion on prey tracking behaviors and the evolutionary arms-race would situate the findings in a broader context and improve the manuscript.

We agree that this is an interesting topic, but there is always a tension between offering an extensive introduction versus getting quickly to the point in a manuscript. Some readers will appreciate the former and others the latter. We aimed for a concise introduction because almost nothing is currently known about the evolutionary history of echolocation and its impact on prey co-evolution. Moreover, it has not been clear to what extent results from other sensory systems apply to echolocating animals. We feel therefore that a review of the broader literature around predator-prey arms races may seem out-of-place to some readers. However, we have added a few words and some additional references on this theme.

Some other comments:Line 52: why do the authors suggest echolocation enables stealthy approaches? It is well documented that a number of prey can hear echolocating animals approaching. This sentence is confusing as it seems to imply echolocation provides more stealth than vision. Please reword.

Thank you for raising this point. We were thinking specifically about marine echolocators here. While many prey species targeted by bats are able to hear echolocation calls, the same is not true for the prey of marine echolocators. Ultrasonic hearing has thus far only been shown in a few species of shad (subfamily: alosinae) where it may have developed in response to predation by delphinids (Wilson et al., J. Exp. Biol. 2009). The typical prey of porpoises, beaked whales and many other toothed whales, to the best of our knowledge, have no such capability (Wilson et al., 2007). For these predators, therefore, echolocation really does provide a private sensory channel. A few bat species may also achieve this benefit, at the expense of detection distance, by producing much weaker calls (Goerlitz et al., Current Biology 2010). We have added text to clarify this point.

Line 73: I suggest you add the citation Salles Diebold and Moss 2020, which also shows an increase in vocalization rate in bats tracking a moving prey item trajectory.

Thank you, we have added this very relevant reference. This excellent study looks at prey-tracking behaviour in bats during regular clicking (i.e., outside of the high-call-rate buzzes that occur during close-in chases) and so provides a broader context to our topic.

Line 83: please correct the sentence to "one in four species of mammals"

Corrected.

Line 212: please clarify that the acceleration of the target moved by hand was calculated with an accelerometer (as described in methods). As it reads it is unclear how controlled the target movements were.

Good point. Amended as suggested.

Line 252: unit typo should read "m.s-3 " or "m/s3 " as in Line 218; not "ms-3".

Thanks. We think the reviewer means line 225 here. In our merged pdf of the submission, this unit appeared correctly but we have checked that it remains correct in the revised submission.

Figure 7: From the figure it looks like the limit to how fast ICI change can occur after prey movement, ~200 ms for captive animals? While in the wild the authors show evidence for 100-200 ms delay after initial prey movement (line 141). Is there a limit to how fast the target is moved by hand that cannot replicate the shorter latencies seen in the wild?

This figure shows the latency to large changes in ICI (which may be comparable to saccades in vision) whereas Figures 2 and 6 show the latency to increases in ICI of any size (i.e., for both proportionate and saccade-like responses). In vision, saccades have longer latency than tracking responses (e.g., Erkelens, Vision Research 2006) and the larger latencies in Figure 7 vs Figures 2 and 6 are consistent with this. However, it is also possible that hand-movements of the target do not reach the highest accelerations achievable by small prey in the wild and therefore do not invoke the fastest responses. This is a topic for further research but we have added a clarification in the text.

Reviewer #2 (Recommendations for the authors):1. Some readers might be confused with the word "whales" in the title, given that this study is based on porpoises too. Perhaps consider changing to "echolocating whales and porpoises"? (or cetaceans?)

It is true that the term 'whales' is imprecise. We have changed this to 'toothed whales' to avoid confusion.

2. In line 53, the authors present the entirely logical and reasonable argument that "prey are likely to detect the predator only when it is close," but because this follows a sentence about vision, it presents the idea-intentionally or unintentionally-that aquatic prey will detect their predators visually. In fact, it seems reasonable that most aquatic prey will use mechanoreception, particularly if a predator approaches from behind swimming prey, and if the prey are fish with sensitive lateral line systems. Even if the beaked whales are preying on squid (which typically rely on visual sensation), this mostly occurs in deep, dark waters where vision is of limited use.

Agreed, and thanks for catching this. We have clarified in the manuscript that mechanoreception is likely the primary modality for prey to detect predators in light-limited marine environments.

3. Is it concerning that the two captive porpoises had similar but really non-overlapping response times ("50-100 ms for Freja and 100-150 ms for Sif"; lines 202-203)?

This is a drawback of captive studies for which few individuals are available. While Freja has shown very consistent echolocation performance over many years of target approach trials, Sif has always been more variable. That both animals showed broadly similar responses over many trials points to the stereotypy of the echo-kinetic response even though the latency varied across the two individuals.

4. Can you provide some additional data about the two captive harbour porpoises (apart from the fact that both Sif and Freja are female; line 383). Size? Age? Length of time in captivity?

Good point. This information is now added.

5. The analogy of a temporal fovea similar to the spatial (retinal) fovea is really nice, and quite evocative.

Thank you. Yes, this has inspired us to think of follow-on experiments to explore how reliant echolocators are on dense temporal flow.

6. The sources supplied by the authors are good and up to date. In line 238, a source should be supplied for the statement about "the prevailing click-by-click model for echolocation processing". Who proposed, stated, or described/explained this model? Would it be Au 1993?

Indeed, citation added.

7. Two related thoughts: First, I have often heard, informally, that cetacean brains are very large but that, unlike most mammals, cetacean brains are largely devoted to processing auditory information. I am not a brain expert by any means, and although I doubt the veracity of this claim, I wonder if the authors might include here, perhaps in the Introduction of Discussion, a brief mention about cetacean brains and their relation to auditory processing abilities.

Although toothed whales do indeed have a well-developed auditory cortex, the large brains in some species may have evolved for different reasons Toothed whales with large relative brain sizes are typically very social species such as pilot whales and bottlenose dolphins that must navigate complex social settings. It is also worth noting that bats with very small brains can perform similar biosonar processing as toothed whales. Although this is a very interesting topic, to do it justice would require a lengthy aside in the manuscript which we think would interrupt the flow.

8. Second, I wonder about subconscious vs. conscious neural processing. I realize that this is not entirely pertinent to this study, but you've got me thinking about it. At the standard frame rate of television and movies (24 fps) we cannot (consciously) distinguish a series of discrete, separate still images-we see them, obviously, as a continuous stream of video input (even if we subconsciously detect subliminal cues in individual frames, for example). Might this relate in any way to this study? I like your two-part model of feedback control (Figure 1) of vocal muscles versus locomotor muscles. Could odontocete brains possibly recognize individual clicks, in the same way that we can "see" individual video frames, even if on another level they show responses that are decoupled from the click rate?

That is an intriguing proposal. Such sub-conscious perception in vision could perhaps be related to differential processing in early neural layers, e.g., to facilitate detection of fleeting predator/prey cues. But there is a big jump from the 24 fps of movies to the 500+ clicks/s of a porpoise and it seems likely that the perception of echo streams from individual clicks would be less valuable to an echolocator than are brief optical cues to a visual predator or prey. Echolocating whales and delphinids are frequently within hearing range of other conspecifics and so must be able to echolocate without being distracted by the clicks of other animals. One important function of the asynchronous processing that we propose in the manuscript may be to improve robustness to such interference, i.e., preventing its perception within the echo processing neural circuitry. However, the presence of conspecifics is also a valuable source of information to an animal (e.g., as in Aguilar de Soto et al., Scientific Reports 2020) pointing to the need for stream segregation in early neural processing.

9. The color figures are crisp and easy to follow, and the captions make sense.10. This manuscript is really very well written (I commend, and thank, the authors for the fine care they have taken in writing this), but an "Oxford comma" is needed in line 44 before the word "or" (third item in a list) to prevent possible confusion.

Thank you for the kind words. The comma is fixed.

References

Aguilar de Soto N, Visser F, Tyack P, Alcazar J, Ruxton G, Arranz P, Madsen PT, Johnson M 2020. Fear of killer whales drives extreme synchrony in deep-diving beaked whales. Scientific Reports, 10:13. DOI 10.1038/s41598-019-55911-3

Erkelens CJ 2006. Coordination of smooth pursuit and saccades. Vision Research 46:163-170. DOI 10.1016/j.visres.2005.06.027

Goerlitz H, Hofstede H, Zeale M, Jones G, Holderied M 2010. An aerial-hawking bat uses stealth echolocation to counter moth hearing. Current Biology 20:1568-1572. DOI 10.1016/j.cub.2010.07.046

Wilson M, Hanlon RT, Tyack PL, Madsen PT 2007. Intense ultrasonic clicks from echolocating toothed whales do not elicit anti-predator responses or debilitate the squid *Loligo pealeii.* Biology Letters 3:225-227. DOI 10.1098/rsbl.2007.0005

Wilson M, Montie E, Mann K, Mann D 2009. Ultrasound detection in the Gulf manhaden requires gas-filled bullae and an intact lateral line. J. Exp. Biol. 212:3422-3427. DOI 10.1242/jeb.033340